# Study of the Impact of Graphite Orientation and Ion Transport on EDLC Performance

**DOI:** 10.3390/ma15010155

**Published:** 2021-12-26

**Authors:** Joseph M. Gallet de St Aurin, Jonathan Phillips

**Affiliations:** 1US Navy, Naval Postgraduate School, GSEAS, Monterey, CA 93943, USA; joegallet@gmail.com; 2Capacitor Foundry, Sand City, CA 93955, USA

**Keywords:** dielectric, oriented graphite, electrolyte, EDLC, ion flow, capacitance

## Abstract

A model study of electric double layer capacitor (EDLC)-style capacitors in which the electrodes were composed of low surface area-oriented flakes of graphite that compressed to form a paper-like morphology has suggested that ion transport rates significantly impact EDLC energy and power density. Twelve capacitors were constructed, each using the same model electrode material and the same aqueous NaCl electrolyte, but differing in relative electrode orientation, degree of electrode compression, and presence/absence of an ionic transport salt bridge. All were tested with a galvanostat over a range of discharge currents. Significant differences in energy and power density and estimated series resistance were found as a function of all the factors listed, indicating that capacitor performance is not simply a function of the electrode surface area. This simple postulation was advanced and tested against data: net ion (Na^+^, Cl^−^) ‘velocity’ during both charge and discharge significantly impacts capacitive performance.

## 1. Introduction

The development of capacitors with high energy and/or power density will help enable the proposed switch from combustion-based energy systems to ‘green’ electric systems that are recharged with renewables. Ideally, and theoretically, capacitor energy density could surpass that of batteries, allowing capacitors to replace batteries. Realistically, it is unlikely that capacitors will ever match the energy density of batteries, but the superior power output and durability of capacitors ensures that capacitors will have a number of niche rolls, including load leveling in systems for which batteries are the primary source (e.g., electric vehicles), battery life extension applications such as for satellites, collecting energy from high power surge sources such as regenerative braking, and providing pulsed power for inherently high power applications, as in [1,2,3].

Given the increased urgency of efforts to create a fully electric power system using only renewable energy sources, resources for research to improve capacitors, particularly electric double layer capacitors (EDLC), also known as Supercapacitors, have increased dramatically in the last decade. In ‘carbon only’ systems, as per this report, virtually all the increased research has been directed at exploring the effect of the microstructure (e.g., graphene, CNT) and the impact of various additives on carbon energy density, power density, and electric conductivity [1,2,4,5,6,7,8,9,10,11,12]. There are some general caveats to much of this research. As pointed out elsewhere, the fact that there are no universal protocols for capacitor testing makes comparisons and claims of superior performance difficult to verify [1,13,14,15,16]. In addition, other factors, including ionic conductivity, fabrication method, absolute electrode dimensions, even measurement set-up, are likely to impact performance [17]. There has been relatively little work designed to quantify the impact of these factors on carbon-based capacitor performance [13,17,18].

The novelty of the present work is the test of the impact of two overlooked factors on EDLC performance: the orientation of graphite planes in the electrodes, and enhanced pathways for ionic transport. Study of these factors show, for the first time, that they dramatically impact energy storage and power production. Regarding orientation, in the absence of any other modification, the orientation of the graphite flake-based electrodes relative to each other led to a factor of >3× change in energy density. Regarding enhancing ion transport, the use of a ‘salt bridge’ to enhance ion transport between electrodes increased both energy density and power production significantly. These results, from the simple ‘model’ system tested, provide experimentally quantitative support for the general postulation that many factors impact EDLC performance.

## 2. Materials and Methods

EDLC Microstructure: The electrodes were created from a commercial material, Grafoil, composed of flakes of graphite compressed to create a ‘paper-like’ material with a very modest surface area [19,20]. In prior work, it was demonstrated that the graphite flake basal planes are strongly oriented parallel to the macroscopic surface of the paper [21].

Two types of Grafoil were employed in capacitor electrodes: (1) Grafoil provided by the manufacture directly; (2) Grafoil mechanically compressed between stainless steel platens to 2000 psi using an Instron SATEC machine (Instron Corp, Norwood, MA, USA). 

One simple measure of the impact of compression was the reduction in measured thickness. The measured thickness of the ‘raw’ material was 0.4 mm, which was reduced to 0.36 mm by compression. To determine the effect of compression on microstructure, both surface area and relative basal plane orientation were determined. A NOVA 4200e Surface Area and Pore Size analyzer (Quantachrome Instruments, Boynton Beach, FL, USA) was used to determine the BET surface area of uncompressed (15.5 m^2^/g) and compressed (12.7 m^2^/g), material. Notably, the measured value for the uncompressed material was similar to that reported by the manufacture, 18.5 m^2^/g, indicating that the measurement technique yields reasonably good quantitative data. In sum, compression at 2000 psi reduces surface area moderately, ~20%.

To test for changes in orientation, anticipated on the basis of earlier studies, relative X-ray diffraction peak heights were determined with a Rigaku MiniFlex 600 benchtop X-ray diffractometer (Rigaku Analytical Devices, Wilmington, MA, USA) (Figure 1). Based on a comparison of intensity ratios between specific crystallographic planes for graphite, it was determined that compression did not modify the relative basal plane alignment of the Grafoil by a statistically significant amount. Both the compressed and uncompressed samples were found to have more than 90% of the graphitic crystals aligned with the macroscopic surface. This is not consistent with earlier work, though might reflect changes in the Grafoil manufacturing process over the decades between studies. That is, the commercial material presently available shows greater alignment of the basal planes and lower surface area than that recorded for the ‘as received’ material in prior decades.

EDLC Electrode Structure: To test the impact of orientation, electrodes were made of Grafoil that was folded to form an accordion structure, in which each fold was 1 cm across and 3 cm long (Figure 2).

Once the surface incisions were made, the Grafoil was folded along these incisions in an alternating pattern such that the incision formed the outer spine of the fold. The opposite side of each initial incision was folded into each “valley”. Each electrode, dry, weighed approximately 1.5 g.

After the accordion-fold was complete, the inside surface of each valley fold was lightly scoured to promote liquid adhesion when the electrolyte was added. Prior to testing of each configuration, two drops of 3% NaCl electrolyte (approximately 0.05 mL/drop) were added to each valley fold on both faces of the electrode. The accordion-fold electrodes were then lightly compressed to close each of the folds.

Capacitor Construction: Following the construction of two Grafoil accordion electrodes, the next step in capacitor construction was to employ an electrically insulating, commercially woven nylon blend separator (84% nylon and 16% spandex) with a measured thickness of 0.4 cm and 50% open space, employed previously in parallel plate capacitors [22], as an electrically insulating, ion transport ‘friendly’ media between two, well oriented, electrodes. To enable ion transport, the woven nylon was wetted with 5 drops of the 3 wt% NaCl solution. In prior work with parallel plate capacitors, this electrolyte was found to qualify as an SDM. Indeed, in earlier work [18,22,23,24], aqueous NaCl solution generally had dielectric constants >10^8^ at low frequency (ca. <1 Hz).

The three ‘specific orientations’ of electrodes that were employed are illustrated in Figure 3. In each configuration, Grafoil surface planes are oriented in a different manner relative to the separator. In the Basal Plane Orientation (BPO), all the macroscopic surfaces of the folded Grafoil are parallel to the separator. Ions diffusing through the separator need to either climb ‘up and over’ a Grafoil layer, or through one, to reach the next valley. In the Edge Plane Orientation (EPO), the Grafoil surface are perpendicular to the separator. Ions diffusing through the separator could directly transport into all ‘valleys’ between the Grafoil sheets. In the Offset Orientation (OO), one electrode is arranged per the BPO, and one per the EPO.

The next step was to create a ‘compressed’, mechanically stable, capacitor by using rubber bands to ‘attach’ the electrodes to a standard sized light microscope slide. In Figure 4, a capacitor in the offset configuration is shown with a ‘salt bridge’. Indeed, each capacitor was tested in two configurations: with and without a ‘salt bridge’. The net was twelve capacitors built and tested, each with a unique arrangement of the materials, orientations, and ‘salt bridge’ level: three different orientations × two different compression levels × two salt bridge configurations (Table 1).

The salt bridge was prepared with a gel prepared by mixing 1.8 g of fumed silica (Sigma Aldrich, 0.007 μm avg. particle size, St. Louis, MO, USA) to which a NaCl (saltwater) solution, 0.6 g NaCl and 20 g DI, was slowly added. Similar gels were previously shown to be effective for ion transport in parallel plate-type capacitors [23,24]. Its use as a salt bridge between electrodes requires the same properties. Additionally, the gel held forms such that it was capable of being shaped. As shown, it was ‘molded’ as a relatively thick layer on one end of the capacitors, creating an avenue for ion transport between the two electrodes.

The final step, to prevent drying, was to place the capacitor, along with a small amount of wetted paper towel, inside a gallon-sized, low density polypropylene bag. Small slits allowed the tabs on the electrodes (Figure 2) to be connected to the galvanostat with alligator clips.

Measurement Protocol: The test apparatus used in each experiment was the BioLogic VSP-300, a multichannel potentiostat/galvanostat. Each capacitor was studied using the same three step /bidirectional constant current method, a protocol similar to that used to characterize commercial capacitors [25]. Step 1: Charge to a selected positive voltage at constant current. Step 2: Hold capacitor at that voltage for a select period. Step 3: Discharge at a given constant current. A cycle is completed by repeating the Steps 1–3, except Step 1 is done to a negative voltage. Each cycle is repeated six times. Next, the discharge current in Step 3 is changed, and a second set of data collected. In total, six cycles, each at a different discharge current, were measured for each capacitor (Table 1). A typical cycle for one capacitor is shown in its entirety in Figure 5.

As noted earlier, there is no generally accepted protocol for permitting claims of superior performance, or even an absolute comparison of any capacitor properties [1,13–16); however, for this study, that is not a significant concern. This study is not about creating a superior capacitor or comparison with prior published reports, but rather designing a methodology which allows for quantitative comparisons for energy storage and power delivery between capacitors with different orientations and different ion transport paths/mechanisms. The method employed undoubtedly serves this purpose well; moreover, it is consistent with the body of research conducted at the Naval Postgraduate School [1,18,22,23,24]. Prediction: any and all alternative approached to capacitive property measurement would lead to the same basic conclusions regarding the impact of graphite plane orientation and ion transport significance.

All techniques for measuring capacitance are imperfect [1]. An illustrative example are the inherent limits of impedance spectroscopy. This technique is generally limited at 300 K to study capacitive behavior below 25 mV, as higher temperatures lead to non-linear behavior. In fact, the fundamental non-linearity of capacitors is reflected in the need to model capacitor behavior with ‘equivalent circuits’, which are often very complex. The selection of constant current discharge for the present study was made because the results, using this approach, for low frequency discharge are the least convoluted, most ‘transparent’, available over wide voltage and current ranges, and comparable to a large body of literature [1,13,14,15,16,18,22,23,24].

Three primary values obtained from the data, specific energy (J/kg), specific power density (W/kg), and capacitance below 1 volt (F/g), are reported here. The reported/plotted specific energy density (J/kg) value for each discharge current was the average of energy density values obtained from the twelve discharge curves (six positive voltage, six negative voltage) collected at that current, divided by the total weight of the dry electrodes. The energy density for each individual discharge curve was computed to be the integrated area under the entire discharge curve (V × sec) multiplied by the discharge current. ‘Power density’ (J/s* kg), a value appropriate for the quantitative comparison of ‘power’ between different capacitors, as established in earlier works [18,23,24], was obtained from each discharge curve by dividing the energy of the discharge by the total discharge time and total electrode mass. The capacitance below 1 volt was computed in the standard fashion [1] for constant current discharge, by dividing the current by the near linear value of the discharge curve (dV/dt). The ESR for each CHD sequence was determined using the industry standard method [25]: dividing the voltage drop ΔV that occurs within the first 10 ms by the constant current discharge rate I. The reported/graphed value at any given discharge current is the average obtained from twelve discharges.

## 3. Results

Overall, the data for the model system showed many of the qualitative results expected for capacitors, including: (1) The higher the discharge current, the higher the power density; (2) The higher the electrode surface area/kg, the higher the energy and power density. Three other results are not anticipated by standard theory, and have not been previously reported: (1) The higher the discharge current, the *higher* the energy density; (2) The geometry of the electrodes significantly impacts both energy and power density (3); A ‘salt bridge’ increases energy and power density. Indeed, the difference in energy density as a function of configuration is virtually eliminated by the salt bridge.

Specific Energy: The specific energies of each of the three configurations with no salt bridge, using either uncompressed or compressed Grafoil, are shown in Figure 6. The electrode configuration clearly impacts performance. BPO configurations resulted in considerably poorer performances than EPO and OO configurations. For example, at a discharge time of 100 s, the uncompressed BPO (~220 J/kg) had a measured specific energy approximately 28% that of the uncompressed EPO (~800 J/kg).

Another clear finding is that, for any particular configuration using compressed Grafoil, electrodes resulted in lower specific energy than its uncompressed counterparts. In the case of the BPO configuration, the difference is very significant. For example, at a discharge time of 1000 s, the compressed BPO electrodes have only about 25 percent the energy density of the uncompressed electrodes. The difference in performance between compressed and uncompressed Grafoil electrodes is significant, but not as dramatic, in the other two configurations. In both the OO and EPO configurations, the uncompressed energy density at a discharge time of 1000 s is <2× larger than that of the compressed electrode configuration.

It is notable that the energy vs. discharge time curves are rather ‘flat’, and, more significantly, have a ‘negative’ slope. That is, for EDLC and other capacitors, the energy delivered to the load generally increases with longer discharge times. In this work, the energy densities decreased as the discharge current was decreased, resulting in an increase in discharge time.

In Figure 7, the impact of the salt bridge on specific energy for the uncompressed Grafoil electrodes in the three configurations is shown. It is evident that the salt bridge increases the energy density in all cases. However, the most important finding is that the salt bridge nearly removes the difference in energy density as a function of configuration. With a salt bridge in place, the three configurations show very similar behavior. Indeed, with the salt bridge in place, the BPO and EPO behaviors are virtually identical and the OO performance is only ~25% better. Finally, it should be noted that the ‘negative slope’ as a function of discharge time is not impacted by a salt bridge.

It is reasonable to compare the capacitance (below 1 volt), and, by inference, energy density, of these electrodes with other pure carbon materials. In earlier studies, a graph of capacitance, as a function of the carbon surface area, demonstrated a linear relationship [2,26]. The slope of the line can be employed to predict capacitance on the basis of the measured surface area. For uncompressed Grafoil with a measured surface area of 15.5 m^2^/g, this suggests a capacitance of 0.93 F/g. As shown in Figure 8, this value is about a factor of two better than that observed in the uncompressed material in the OO configuration. The figure also shows that comparisons of ‘capacitance’ are fraught with confusion, as so many parameters, such as voltage range, over which capacitance is measured, measurement frequency, ion identity, and details of construction, which are not simply carbon surface area, impact this value. In sum, the results of this study are qualitatively consistent with the anticipated capacitance based on earlier correlations. That is, the energy density per unit area of the electrodes is in close quantitative agreement with earlier findings.

Specific Power: The specific power (W/kg) for the six capacitors, made with the uncompressed Grafoil, all three configurations with and without salt bridge, are shown (Figure 9). For the three capacitors without a salt bridge, as with the energy density results, the power density is clearly a function of the configuration. The BPO configuration without a salt bridge delivers the least power. For example, for a 1000 s discharge, it would deliver ~2.5 × 10^−1^ W/kg, which is about 30% of the power delivery (~8.0 × 10^−1^ W/kg) for the uncompressed EPO configuration.

As with the case for energy density, the power density differences collapse for the capacitors built with salt bridges. All three configurations with a salt bridge have very similar power density curves as a function of discharge time. Finally, it is notable that the power density curves are of standard slope. That is, as the discharge times decrease, and the delivered power increases.

Estimated Series Resistance: The finding that the energy and power density values of all the capacitors studied herein increases with shorter discharges may reflect the unusual ESR behavior. As shown in Figure 10, the ESR values are not constant, but rather a function of the discharge time/current. For all the capacitors, compressed and uncompressed, with and without a salt bridge, ESR values are higher for smaller currents/longer discharges. Given that the load and the output resistance form a voltage divider, increased output resistance will concomitantly reduce the energy drop across the load. Hence, the ESR trend with the discharge time is both consistent with, and explains, the observed drop in the delivered energy at longer discharge times.

The ESR data show clear trend lines for any particular capacitor, but certainly do not suggest a clear quantifiable formula linking ESR to orientation, the state of compression, or presence of a salt bridge. Table 2 does suggest some weak qualitative correlations. First, for any given orientation, compression increases ESR, and second a salt bridge reduces ESR. It is also clear that there is not a strong orientation dependence. For example, all three configurations are the same with compressed/no salt bridge electrodes within experimental error. Uncompressed/salt bridge capacitors are all essentially identical.

Ragone Plots: Presenting the data in Ragone format helps illustrate the unique behavior of all these capacitors (Figure 11). In particular, plotting the data in the standard Ragone format shows that the ‘slopes’ are inverted. In general, Ragone charts show that, for both batteries and capacitors, delivering more power comes at the expense of reducing the energy delivered. Thus, the lines of conventional capacitors and batteries, plotted as per Figure 11, show a negative slope. For all capacitors studied herein, there is no trade-off. Both energy and power increase as the discharge time is reduced, yielding positive slopes.

The power delivered curves also show a remarkably high slope. This reflects the fact that, unlike standard capacitive behavior, the energy delivered increases as the discharge time is reduced. The power delivered is the ratio of the energy delivered to the time of delivery. Conventionally, the numerator term decreases with shorter discharge time. Still, typically, this value does not decrease as rapidly as the denominator, hence the power increases with the decreasing discharge time/higher current. For the capacitors studied herein, the numerator increased, and the denominator decreased, leading to a sharp slope.

## 4. Discussion

This study was conducted to determine, on a fundamental level, whether either or both orientation and ion transport ‘velocity’ can impact the performance of EDLC capacitors. In order to minimize the number of complicating factors, the electrodes were created with low surface area, pure graphite flakes, strongly oriented with basal planes parallel to the surface.

The study, as intended, produced clear empirical findings regarding the target questions of the impact of electrode orientation and the potential impact of ion transport. Regarding electrode orientation: As shown in Figure 6, the orientation can change the energy density by a factor of more than 3×, and, as shown in Figure 8, the orientation can impact the power density by more than a factor of 2. Regarding ion transport: In all cases, the use of a salt bridge increased performance, although the magnitude varied as a function of electrode orientation. Capacitors with electrodes in orientations that yielded higher energy and power density even without salt bridges were marginally improved. The orientation with the lowest energy and power density without a salt bridge, BPO, improved significantly. In net, using a salt bridge effectively made all electrode orientations roughly equivalent performers. Postulate: Ion transport in EDLC can be performance determining. Note: It is understood that improved ion transport only improves performance to a finite limit.

Other data supports the suggestion that ion transport, and not always electron transport, can be performance limiting. For example, it was found that compression of the Grafoil led to a 25% reduction in surface area, but far larger fractional reductions in energy and power density in all cases. Indeed, the reduction was as great as an order of magnitude decrease in these values. Yet, compression, logically, should increase electron transport by enhancing contact between graphitic plates and concomitantly reduce ion transport by ‘shrinking’ or eliminating channels for ion transport. Thus, the large reduction in performance to Grafoil compression can logically be attributed to reductions in ion transport, but not to a reduction in the electron conductivity of the electrodes.

Can the velocity of ion transport somehow explain the inverted slope for the energy vs. discharge time curves (Figure 6, Figure 8 and Figure 10)? Postulate: The energy and voltage of the charged species, electrons, on the electrodes are higher if more ‘ionic’ dipoles (Na^+^ or Cl^−^) are present in the electric double layer. This concept is a variation on the recently postulated Theory of Superdielectric Materials (T-SDM), as discussed elsewhere [27,28,29]. Thus, given a constant charging period, a system with faster ion transport will allow more ions to travel and ‘add’ to the electric double layer than a system with lower ion transport rates. Conversely, this same ion transport ‘advantage’ can become disadvantageous the greater the discharge time. For lower discharge currents/longer discharge times, more ions ‘retreat’ and neutralize (e.g., Na^+^ + Cl^−^ => NaCl). This reduces the number of electric dipoles in the boundary layer, and hence reduces the energy and voltage of the charges remaining on the electrodes. This implies that, the faster the electrons are removed, the fewer ions, which clearly move much more slowly than electrons, will neutralize. Consequently, rapid discharge should lead to more net energy falling on the load, as observed.

In order to gain additional insight into the impact of various factors on ‘transport’, the ESR was measured for all capacitors. The impact of a salt bridge on the ESR, an addition to the capacitor which should only impact ion transport, is revealing. In all cases the salt bridge reduced the ESR value. This trend raises this question: why should ion transport change net resistance? Is not net resistance a function of electron transport? A postulated answer: Ion and electron transport are ‘coupled’. If ion transport is enhanced, so too is electron transport. The physics behind this proposed coupling is not obvious, or is at least ‘complicated’, and was not be considered here.

## 5. Conclusions

In conclusion, the key empirical finding is that, in carbon-based EDLC, enhanced ion transport in the electrolytic material improves performance. Increasing ion transport rates via the proper orientation of electrodes, the inclusion of a salt bridge, or an increased porosity also increases energy and power density. Moreover, it appears that ion and electron transport are linked. If ion transport is enhanced, electron transport is as well. The empirical findings are certain, and a simplistic model of improved performance linking to ion transport enhancement appears to be at least consistent with all observations. Yet, the underlying physics is not clear. Why are ion transport and electron transport coupled? As the ions retreat/neutralize, does this reduce the energy of the charges remaining on the electrodes?

## Figures and Tables

**Figure 1 materials-15-00155-f001:**
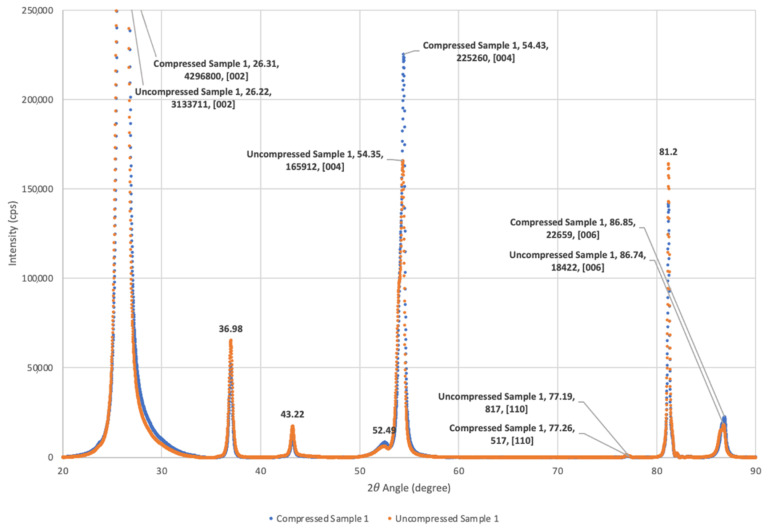
*Compressed and Uncompressed Grafoil XRD*. Both compressed and uncompressed XRD show high orientation of [002] graphite basal planes parallel to surface. For the true powder pattern diffraction pattern arising from ground Grafoil, the [110] (77.19° 2θ) and [112] (81.2° 2θ) diffraction lines are nearly equal in intensity. In contrast to the powder pattern, for both the compressed and uncompressed materials employed in this study, the [110] peak is nearly invisible, but the [112] peak is quite strong [21]. (Note: Only ~6% of the [002] reflection is shown in order to allow the relative intensity of the other peaks to be clearly shown).

**Figure 2 materials-15-00155-f002:**
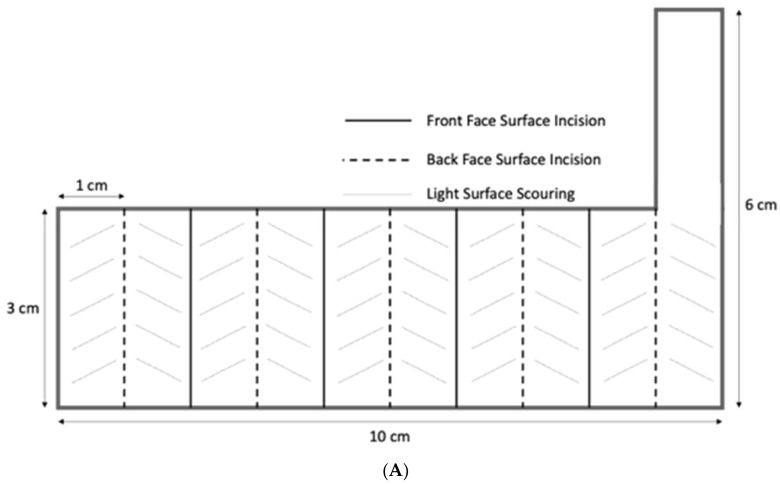
*Accordion Electrode Structure*: (**A**) Dimensions shown. For each electrode, a section of Grafoil, 10 cm × 3 cm + a tab at one end, was cut, scored, and folded as shown. (**B**) After initial folding, ~0.01 cm^3^ of 3% NaCl/DI electrolyte was added to each ‘valley’. (**C**) The completed electrode.

**Figure 3 materials-15-00155-f003:**
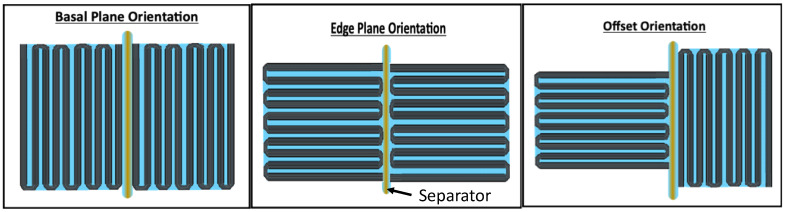
*Edge View of the Grafoil Sheet Orientation.* Capacitors were constructed using accordion folded Grafoil with three ‘specific orientations’ relative to the separator, as illustrated.

**Figure 4 materials-15-00155-f004:**
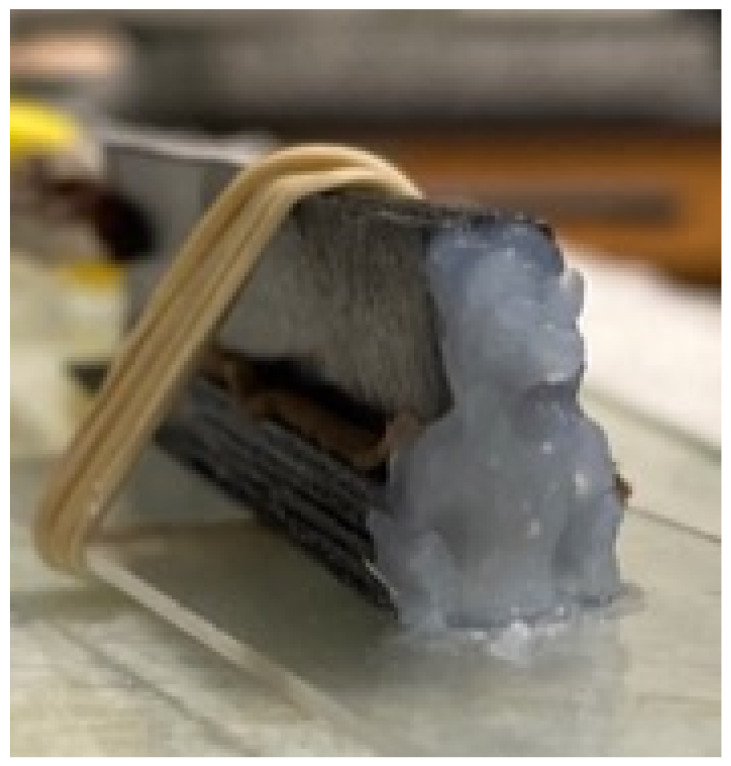
*Salt Bridge*. Capacitor in offset configuration with molded ‘salt bridge’ at one end.

**Figure 5 materials-15-00155-f005:**
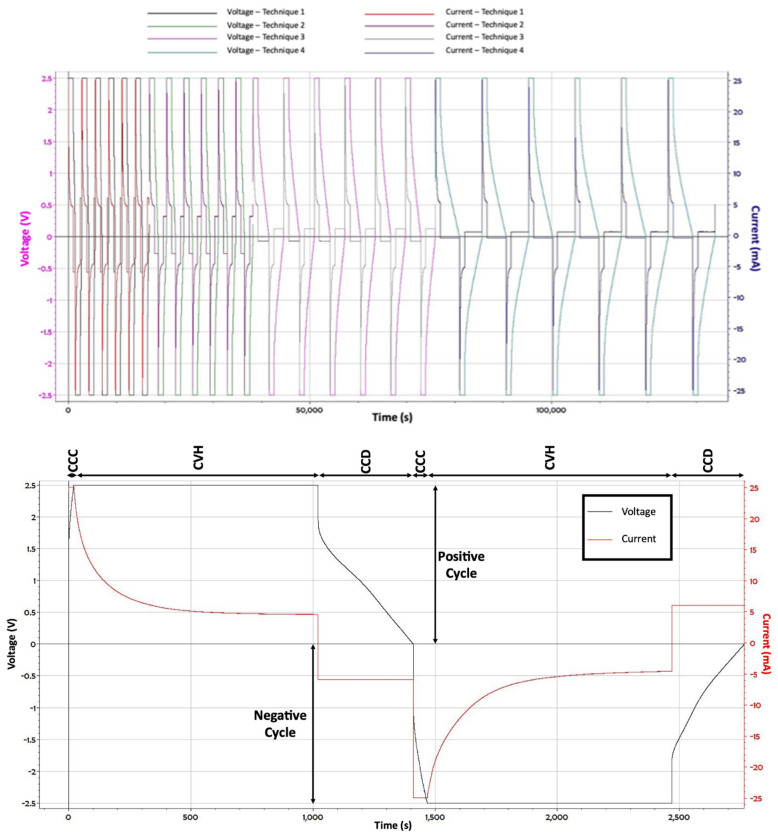
*Example Full Data Set for One Capacitor:* Top: All the data from one capacitor: four discharge currents and six cycles of each (Table 1). Bottom: An expansion of data showing a single cycle. Note: Theoretically, for a capacitor of constant capacitance, the discharge is linear with time. For these capacitors, that was approximately true below ~1.9 Volts. Above that value, voltage dropped sharply, and little energy was delivered to the load.

**Figure 6 materials-15-00155-f006:**
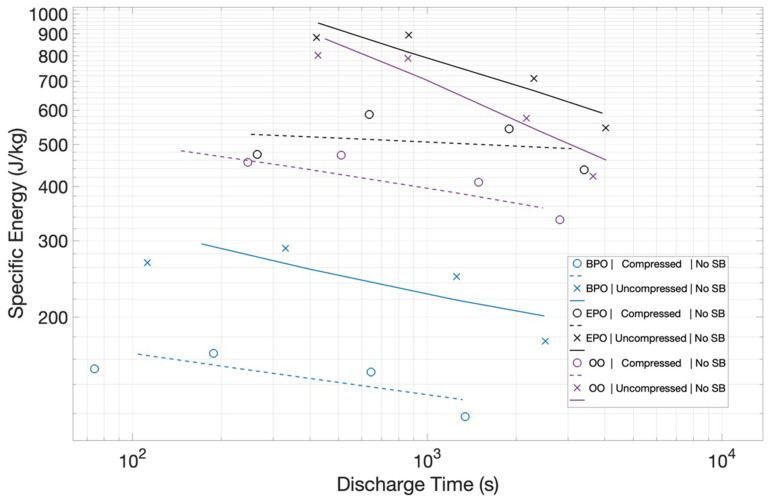
*Summary of Specific Energy for Salt Bridge Free Capacitors.* The energy density data for the six capacitors is shown. One previously unreported trend is apparent: The energy density is a strong function of electrode orientation. It is also clear that compressing/reducing the surface area of the graphitic electrode material reduces specific energy, and energy density decreases slowly with decreasing discharge time.

**Figure 7 materials-15-00155-f007:**
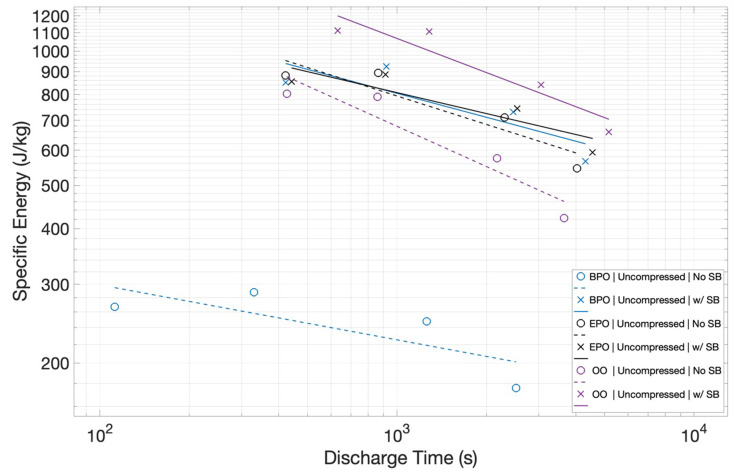
*Impact of Salt Bridge on Specific Energy*. The use of a salt bridge (solid lines) improves performance for all three configurations. More significantly, it ‘collapses’ the differences in energy density as a function of configuration.

**Figure 8 materials-15-00155-f008:**
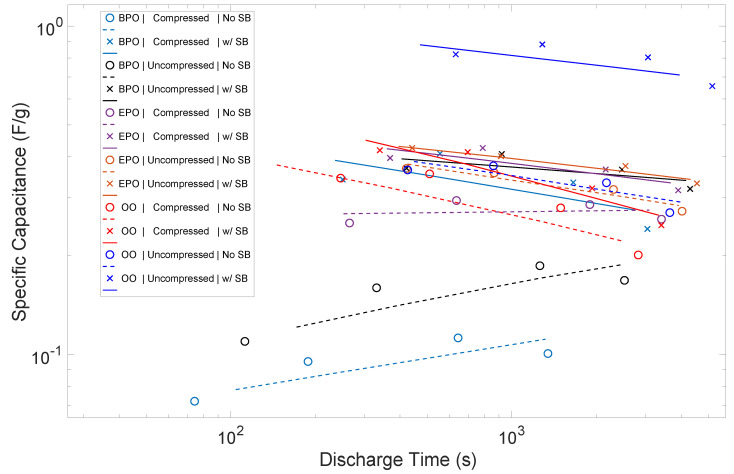
*Capacitance as a Function of Discharge Time*. Consistently, the presence of a salt bridge (solid lines) improves capacitance, all other factors were unchanged.

**Figure 9 materials-15-00155-f009:**
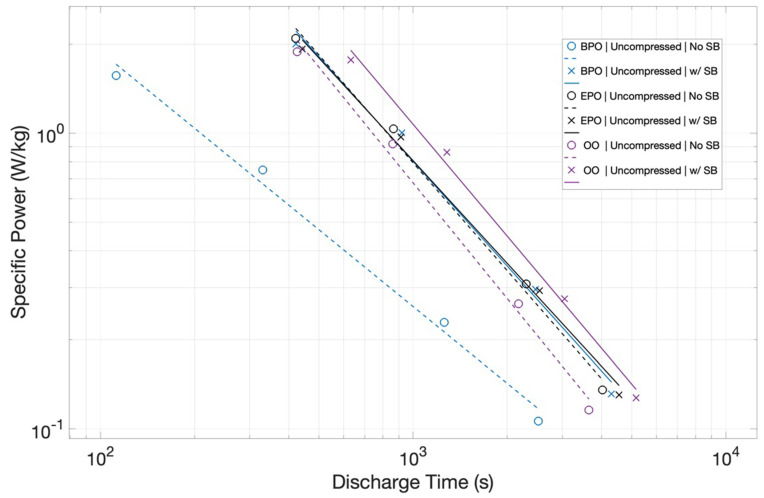
*Specific Power Increases with Higher Current/Lower Discharge Time*. Configuration impacts power significantly in the absence of a salt bridge, but the specific power value differences are almost independent of configuration when a salt bridge (solid lines) is added.

**Figure 10 materials-15-00155-f010:**
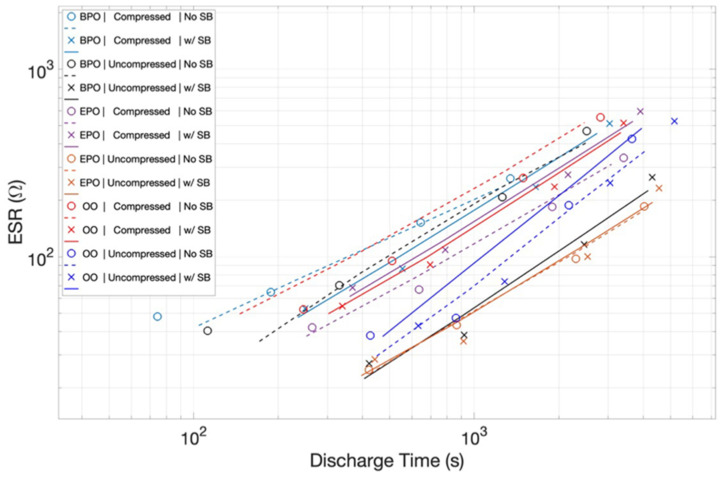
*ESR of all capacitors as a Function of Discharge Rate.* It is notable that ESR is a strong function of discharge time for all capacitors studied.

**Figure 11 materials-15-00155-f011:**
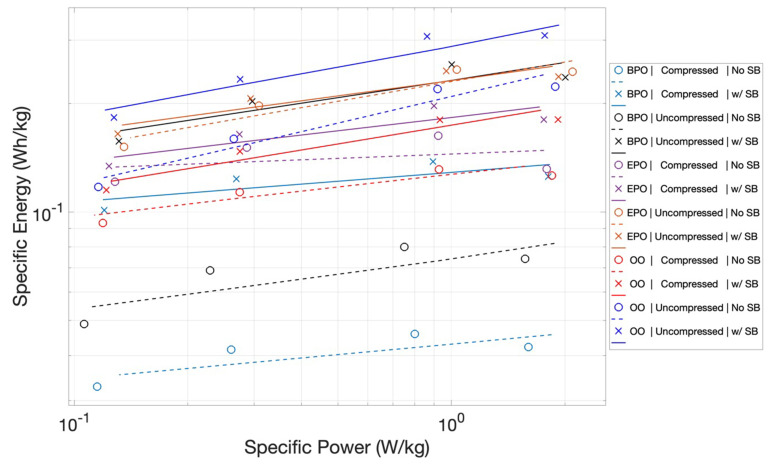
*Ragone Plot*. These plots show the same configuration and salt bridge impacts observed in the specific energy and power plots. Also notable is that the plots are all positive slopes, whereas, in general, for capacitors, the slopes on a Ragone chart are negative.

**Table 1 materials-15-00155-t001:** Summary of Construction and Testing Protocol for 12 Capacitors.

NPS Configuration			Experiment Protocol						
Electrode	Salt Bridge?	Orientation	Sequences per Technique	CCC	CVH	CCD Technique			
						1	2	3	4
Uncompressed Grafoil	No Salt Bridge	BPO	6 Positive 6 Negative	25 mA	2.5 v, 1000 s	6 mA	3 mA	1 mA	0.5 mA
		EPO							
		OO							
	With Salt Bridge	BPO							
		EPO							
		OO							
Compressed Grafoil	No Salt Bridge	BPO							
		EPO							
		OO							
	With Salt Bridge	BPO							
		EPO							
		OO							

CCC—Constant current charge. CVH—Constant voltage hold. CCD—Constant current discharge. (Illustrated, Figure 4, bottom.).

**Table 2 materials-15-00155-t002:** ESR as a function of orientation, compression, and salt bridge.

ORIENTATION	STRUCTURE	RESISTANCE, Ohms (@1000 s)
**BPO**	Uncompressed/No salt bridge	150
BPO	Compressed/No salt bridge	200
BPO	Uncompressed/Salt bridge	30
BPO	Compressed/Salt Bridge	190
**EPO**	Uncompressed/No salt bridge	60
EPO	Compressed/No salt bridge	200
EPO	Uncompressed/Salt bridge	30
EPO	Compressed/Salt Bridge	100
**OO**	Uncompressed/No salt bridge	70
OO	Compressed/No salt bridge	210
OO	Uncompressed/Salt bridge	30
OO	Compressed/Salt Bridge	25

## Data Availability

A great deal of the source data can be found here: Joseph Gallet de St. Aurin, M.S. Thesis: FOLDED CARBON ELECTRODES: A NOVEL APPROACH TO ENHANCING SUPERCAPACITOR PERFORMANCE Naval Postgraduate School. 2021.

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
