# Peer review of "Study of the Impact of Graphite Orientation and Ion Transport on EDLC Performance"

_materials, 2021, doi:10.3390/ma15010155_

Round 1

Reviewer 1 Report

The paper is quite well written and the concept of the experimental design is clever. Major revisions are required for some reasons as described.

  1. While the writing is great, the finer editing needs work. There are a lot of random spaces. The reference list formatting needs to be consistent. There are a couple of errors like on line 161 "...computed to be value was...". Some might consider there to be an overuse of 'speech marks'.
  2. Most of the figures are below acceptable resolution. In particular, figures derived from EC-lab should get their axes re-labelled to be legible without needing to be zoomed in on.
  3. The authors reference Stoller et al and their work on best practice methods for testing, as well as passing comment on the lack of standardized methods, but then appear to choose their own arbitrary and non-discussed / unjustified approach. Why? The remaining points mostly highlight particular concerns with this.
  4. There is no mention of the electrolyte solvent (except for in the salt bridge) but presumably it is water. If this is the case, the potential range for testing seems highly inappropriate without justification. I.e. there is no reason to think that NaCl should suppress the onset of electrolysis of water at ~1.2V. To choose 2.5V is extreme. (Stoller et al recommends 1V)
  5. Why is the presentation of results oriented towards energy density and not capacitance of the material/s? (Stoller et al says capacitance is the most important value)
  6. Exactly how some values are calculated is a bit unclear. E.g. line 162 mentions how energy density is calculated but omits how/or what value of mass or volume the device energy is normalized to.
  7. Which section of "area under the discharge curve" is used for calculations? 
  8. What is the justification for reversing the polarity of the device after initial polarization? In general, this opens up a range of possible issues that may affect the results, but is particularly concerning given the large potential range and the likelihood of nascent H2 or O2 evolution (and possibly carbonates and sodium salts) within the electrodes. 
  9. Why are charge currents not matched to the discharge currents? 
  10. Why is there results/discussion around the metric of discharge time rather than discharge current, since the former seems dependent on the latter? I.e. the experimental protocol in table 1 presents discharge currents for techniques 1-4, but then the results and discussion leave the reader to try to interpret how these translate to discharge times for techniques 1-4. Times which, incidentally, appear to have very neat numbers like 100s or 1000s. Perhaps this reviewer has misunderstood the experimental protocols and results... 
  11. Lines 203 - 206 demonstrate some discussion that is a bit unclear. How are the discharge curves both "rather 'flat'" AND "have a 'negative' slope"? Also, how appropriate is it to describe trends as "flat" on a log-axis?
  12. Lines 66-68. Determining surface area by BET model fitting to gas sorption data is notoriously misapplied / unreproducible at research levels. It would be great to include the necessary information for reproducibility. I.e. sample size, degas conditions, p/p0 range for fitting, or statement that some standard method was used (e.g. ASTM-D8325 or similar). Be cautious about asserting that the "similar" result to the manufacturer spec confirms a good approach, but yet the (absolute) difference between your uncompressed and compressed results is lower than the above "similar" results. How can the first difference be low enough to confirm the approach is good, while the second smaller difference is significant? 
  13. Lines 71-80. A reference and/or data for this part seems appropriate given it is the evidence for the concept that the study is based on. I.e. orientation of graphitic domains on the microscale. It makes it less easy for a critic to argue that the differences you observed between devices are related electrolyte access to the macroscopic valley channels. 

While there is a lot of feedback here, I just want to reemphasize that this work is based on a clever line of inquiry towards simplifying an EDLC system for gaining some fundamental, empirical insight. It really just needs a thorough polish up and/or a little retesting, depending on the authors justification for their non-standard testing approaches. 

Reviewer 2 Report

This paper studied the influence of electrode configuration and the salt bridge on the energy storage performance of a supercapacitor. There are a few questions that should be addressed.

  1. What is the motivation to study the influence of the electrode configuration?
  2. Some important parameters were studied, such as specific energy, specific capacitance, specific power, and ESR. However, the relations among those parameters are not clear.
  3. Following the above question. Although it was found that the electrode configuration and the salt bridge show great influence on the energy storage performance, the underneath mechanism has not been fully discussed.
  4. The resolution of the figures is very low. It is very hard to read the data.

In summary, this manuscript needs major revision before acceptance for publication.

Reviewer 3 Report

In this research article, good scientific accomplishments are provided that are in the study of EDLC capacitors in which the electrodes are composed of low surface area, oriented flakes of graphite, compressed to form a paper like morphology, suggests that ion transport rates significantly impact EDLC energy and power density. The authors reported the fabrication process of the capacitors and demonstrated the performances of the capacitors by investigating different characterizations. However, most of the measurements and analyses are not well organized and explained. The manuscript would not meet the standard for the publication on this journal in present form. The manuscript should be well revised as indicated below in order to eliminate the shortfall and improve the scientific quality of this paper before accepting for publication:

  1. English should be revised to avoid any grammatical mistakes.
  2. Authors are advised to include the novelty of this work.
  3. Authors should provide the XRD, BET, CV and EIS curves in the revised manuscript.
  4. Please include the explanation why they used NaCl electrolytes rather than other electrolytes.
  5. To check the stability of the device, authors are advised to measure 10000 cycles GCD at a constant current and EIS measurements before and after 10000 cycles GCD.
  6. All figures should be revised. It is very difficult to read.
  7. Authors should provide how they calculated the capacitance, Energy density, Power density and resistance values in revised manuscript.
  8. The conclusion part must be revised to reflect the aim of this work.
  9. These few recent publications could be informative for authors and suggested to cite in introduction part of this paper. Energies 2021, 14, 4250; Materials 2021, 14, 2942; ACS Appl. Mater. Interfaces 2017, 9, 10730−10742.

Round 2

Reviewer 1 Report

The authors have not addressed concerns regarding the scientific soundness of this work.

  1. Aqueous devices should not be cycled above 1.2V without discussion and justification. 
  2. Device polarity should not be reversed without discussion, justification and evidence that it is needed/beneficial in the context of an energy storage device.
  3. Science should be reproducible. To promote this, if experimental details are requested, they should be supplied when requested, to the best of the experimenters capability to do so.

Author Response

Basic Response-  The reviewer is making impossible demands.  This is a 'Wizard of Oz' reviewer. We could bring the reviewer everything demanded and then he would simply invent new and even more unreasonable demands. 

  1. Aqueous devices should not be cycled above 1.2V without discussion and justification.        The protocol we employed has been employed in  more than eight publiished papers.  We have never encountered this criticism. The other reviewers did not make it either!  So, we feel there is no basis for not repeating that protocol.   Note: We employed voltages about 1.2 V, yet the world did not end...nor was any change in behavior over many cycles observed.  So, why not?  Why not? Passive aggressive rule making?
  2. Device polarity should not be reversed without discussion, justification and evidence that it is needed/beneficial in the context of an energy storage device.  ....Same response as 1.  And, we believe the polarity should ALWAYS be reversed to demonstrate chemistry is not responsible for observations. I will start rejecting papers that don't follow my rules.  Indeed, as the author of more than 150 reviewed publications and 40 issued US patents, I proclaim myself unchallengable expert. 
  3. Science should be reproducible. To promote this, if experimental details are requested, they should be supplied when requested, to the best of the experimenters capability to do so....Same response as 1.  The reviewer will have to explain precisely what is missing.  We gave an enormous amount of detail.

Reviewer 2 Report

The authors addressed all questions. This version can be accepted for publication.

Reviewer 3 Report

The authors have modified the revised manuscript according to the reviewers comments. Overall, this revised manuscript is suitable for the publication on this journal.

Author Response

Thank you.